# Simultaneous Determination of Four Marker Compounds in *Lobelia chinensis* Lour. Extract by HPLC-PDA

**Beom-Geun Jo** [1,†], **Young-Hun Park** [1,†], **Ki Hyun Kim** [2] , **Su-Nam Kim** [3,*] **and Min Hye Yang** [1,*]

[1] Department of Pharmacy, College of Pharmacy, Pusan National University, Busan 46241, Korea; bg_jo@pusan.ac.kr (B.-G.J.); pyh4061@pusan.ac.kr (Y.-H.P.)

[2] Natural Product Research Laboratory, School of Pharmacy, Sungkyunkwan University, Suwon 16419, Korea; khkim83@skku.edu

[3] Natural Products Research Institute, Korea Institute of Science and Technology, Gangneung 25451, Korea

\* Correspondence: snkim@kist.re.kr (S.-N.K.); mhyang@pusan.ac.kr (M.H.Y.); Tel.: +82-33-650-3503 (S.-N.K.); +82-51-510-2811 (M.H.Y.); Fax: +82-33-650-3419 (S.-N.K.); +82-51-513-6754 (M.H.Y.)

† These authors equally contributed to this work.

**Abstract:** *Lobelia chinensis* Lour. (*L. chinensis*) has traditionally been used as a treatment for snake bites, high fever, jaundice, edema, and diarrhea, and modern studies have reported its anti-inflammatory, antioxidant, and antiviral activities. *L. chinensis* contains various compounds, such as flavonoids and coumarins, and its flavonoid components have been identified in many studies. In this study, a high-performance liquid chromatograph equipped with a photodiode array (PDA) detector and an Aegispak C18-L reverse-phase column (4.6 mm × 250 mm i.d., 5 μm) was used to simultaneously analyze four marker components in *L. chinensis* for standardization purposes. HPLC-PDA (detection at 340 nm), performed using a 0.1% formic acid-water/0.1% formic acid-acetonitrile gradient, separated the four marker compounds: luteolin-7-*O*-β-D-glucuronopyranosyl (1→2)-*O*-β-D-glucuronopyranoside, clerodendrin, chrysoeriol-7-*O*-diglucuronide, and diosmin. The developed analytical method showed excellent linearity values ($r^2 > 0.9991$), limits of detection (LODs: 0.376–2.152 μg/mL), limits of quantification (LOQs: 1.147–6.521 μg/mL), intra- and inter-day precisions (RSD < 1.96%), and analyte recoveries (96.83–127.07%; RSD < 1.73%); thus, it was found to be suitable for the simultaneous analysis of these four marker compounds in *L. chinensis*.

**Keywords:** *Lobelia chinensis* Lour.; luteolin-7-*O*-β-D-glucuronopyranosyl (1→2)-*O*-β-D-glucuronopyranoside; clerodendrin; chrysoeriol-7-*O*-diglucuronide; diosmin; HPLC-PDA

## 1. Introduction

*Lobelia chinensis* Lour. (*L. chinensis*), a perennial herb of the Campanulaceae family, is distributed mainly in Indochina, Taiwan, Japan, China, and Korea. It is popularly known as "Aze-mushiro" or "Mizo-kakushi" in Japan, "Ban-bian-lian" in China, and "Su-yeom-ga-rae-kkot" in Korea. Dried whole *L. chinensis* is widely used in traditional Chinese medicine (TCM). According to the records of the TCM book "Ben Cao Gang Mu", it has been used to treat snake bites, diarrhea, jaundice, edema, and high fevers due to malaria [1]. In addition, pharmacological studies have reported that *L. chinensis* has anti-inflammatory [2], anti-oxidative [2,3], anti-viral [4,5], anti-obesity [6], anti-tuberculosis [7], anticancer [8], and antitumor [9,10] activities.

The chemical components isolated and identified were flavonoids (luteolin, apigenin, apigenin 7-*O*-rutinoside, diosmin, diosmetin, linarin, wogonoside, 3′-methoxyl-linarin, and lobelitin A-G), alkaloids (lobeline, norlobelanine, lobelanine, lobechinenoids A-D, lobechidine A-C, 8,10-diethyllobelidione, 8,10-diethyllobelidiol, 8-propyl-10-ethyllobelionol, and 8-ethyl-10-propyllobelionol), coumarins (6,7-dimethoxycoumarin, 5-hydroxy-7-methoxycoumarin, 5,7-dimethoxy-6-hydroxy-coumarin, scoparone, and citropten), terpenoids (phytol, phytenal, cycloeucalenol, and 24-methylene-cycloartanol), and polyacetylenes (lobetyolinin, lobetyolin, and isolobetyol) [2,4,8,11–15]. Of the compounds in *L. chinensis*,

diosmin, diosmetin, linarin, lobetyolinin, and lobetyolin are considered to be the main active components [8,12,13]. Biologically, the flavonoids diosmin, diosmetin, and linarin have been confirmed to have hepatoprotective [16,17], anti-inflammatory [18–23], anti-oxidative [19,24–26], and anti-atopic [27,28] activities, and the polyacetylene lobetyolin has been reported to have anti-inflammatory [29,30] and antioxidant [31] activities.

In previous studies on *L. chinensis*, pattern analysis, chemical profiling, and qualitative and quantitative analysis have been performed [32,33]. Using high-performance liquid chromatography combined with a diode-array detector and coupled with electrospray ionization with ion-trap time-of-flight mass spectrometry (HPLC-DAD-ESI-IT-TOF-MS), 11 compounds were identified from *L. chinensis* by comparing their retention times and MS spectra with those of standards or literature data [32,33]. However, no report has yet been issued on the simultaneous multicomponent analysis or quality evaluation of *L. chinensis* using HPLC-PDA. The formulation of systematic quality evaluation criteria is necessary because qualitative and quantitative analysis results depend on varieties, cultivation conditions, and harvesting seasons used. Therefore, in this study, for quality control purposes we developed and validated a simultaneous analysis method for luteolin-7-*O*-β-D-glucuronopyranosyl (1→2)-*O*-β-D-glucuronopyranoside, clerodendrin, chrysoeriol-7-*O*-diglucuronide, and diosmin (major components of *L. chinensis* extract) using HPLC-PDA (high-performance liquid chromatography performed using a photodiode array detector).

## 2. Experimental

### 2.1. Plant Material

Whole *Lobelia chinensis* Lour. (Campanulaceae) plants were collected from The Institute of Medicinal Plants at Kolmar BNH Co., Ltd. (Jecheon, Chungbuk Province, Korea) in November 2020 and authenticated by Hyuk Joon Kwon (Ph.D. in Agriculture, The Institute of Medicinal Plants at Kolmer BNH Co., Ltd.). A voucher specimen (PNU-0036) was deposited at the College of Pharmacy, Pusan National University, Busan, Republic of Korea.

### 2.2. Chemicals and Reagents

The diosmin (DSM, purity ≥ 98%) reference compound was purchased from Chem-Faces (ChemFaces Biochemical Co., Ltd., Wuhan, China). Luteolin-7-*O*-β-D-glucuronopyra nosyl (1→2)-*O*-β-D-glucuronopyranoside (L7GC, purity ≥ 98%), clerodendrin (CDR, purity ≥ 96%), and chrysoeriol-7-*O*-diglucuronide (C7dGlu, purity ≥ 98%) were isolated from *L. chinensis*, identified in our laboratory, and used as reference compounds [34–36]. The chemical structures of the four reference compounds are shown in Figure 1. For the HPLC-PDA analysis, HPLC-grade water and acetonitrile were purchased from Honey-well Burdick & Jackson (SK Chemicals, Ulsan, Korea) and HPLC-grade formic acid and dimethyl sulfoxide (DMSO) were purchased from Daejung Chemicals (DAEJUNG Chemicals & Metals Co., Ltd., Siheung-si, Korea) and Junsei Chemical (Junsei Chemical Co., Ltd., Tokyo, Japan), respectively.

### 2.3. Equipment

HPLC was performed using the Waters Alliance e2695 system (Waters Corporation, Milford, MA, USA) equipped with a 2998 photodiode array (PDA) detector and an Aegispak C18-L column (4.6 mm × 250 mm I.D., 5 μm, Young Jin Biochrom Co., Ltd., Seongnam, Korea).

**Figure 1.** Chemical structures of four compounds (luteolin-7-*O*-β-D-glucuronopyranosyl (1→2)-*O*-β-D-glucuronopyranoside, L7GC; clerodendrin, CDR; chrysoeriol-7-*O*-diglucuronide, C7dGlu; diosmin, DSM).

### 2.4. Chromatographic Conditions

Chromatographic analysis was performed using a C18 column maintained at 30 °C. The mobile phase used for the chromatographic separation consisted of (A) 0.1% formic acid–water (*v/v*) and (B) 0.1% formic acid–acetonitrile (*v/v*), and gradient elution was performed using a linear gradient of 15–20% (B) over 0–5 min; 20–22% (B) over 5–11 min; 22–24% (B) over 11–15 min; 24–30% (B) over 15–20 min; 30–55% (B) over 20–23 min; isocratic elution with 55% (B) over 23–29 min; and then returned (29–29.01 min) to 15% (B). The flow rate used was 1.0 mL/min, and the sample injection volume was 10 μL. The detection wavelength range was set from 190 to 400 nm, and data collection and processing were performed using the Empower version 3 software (Waters Corporation, Milford, MA, USA).

### 2.5. Preparation of Crude Extracts and Sample Solutions

Cold, air-dried (29–31 °C), whole (1 g) *L. chinensis* was extracted twice by heat reflux extraction at 90 °C for 3 h in 10 mL of distilled water. After paper filtration (Advantec No. 2, Tokyo, Japan), the extract was concentrated in vacuo and freeze-dried to obtain the powder extract.

Sample solutions (concentration: 20 mg/mL) were prepared by accurately weighing powder extract and dissolving it in water. This solution was filtered through a 0.45 μm PTFE syringe filter (13HP045AN, Advantec, Tokyo, Japan) before being injected into the HPLC system.

### 2.6. Preparation of Standard Solutions

Stock standard solutions of L7GC (concentration: 900 μg/mL), CDR (concentration: 1800 μg/mL), C7dGlu (concentration: 900 μg/mL), and DSM (concentration: 1500 μg/mL)

were prepared by accurately weighing standards and diluting it in DMSO-methanol-water solutions (5:4:1, *v/v/v*). Working standard solutions were prepared by dilution in DMSO-methanol-water solutions (5:4:1, *v/v/v*) at the appropriate concentration ranges specified for validation, then filtered through a 0.45 μm PTFE syringe filter (13HP045AN, Advantec, Tokyo, Japan) before HPLC injection.

### 2.7. Validation of the HPLC-PDA Method

The simultaneous analysis method was validated as described by the International Conference on Harmonisation (ICH) guidelines [37] for specificity, linearity, limit of detection (LODs), limit of quantification (LOQs), precision, analyte recovery, and solution stability.

Specificities were determined to confirm that L7GC, CDR, C7dGlu, and DSM were selectively separated from other compounds in the crude extract and determined using the retention times and absorbance wavelengths of the chromatograms obtained by analyzing samples and standard solutions with HPLC-PDA.

To assess linearity, stock solutions of L7GC, CDR, C7dGlu, and DSM were diluted with six target concentration ranges (1%, 10%, 50%, 80%, 100%, and 120%) to obtain L7GC and C7dGlu at six target concentrations—i.e., 3.00, 30.00, 150.00, 240.00, 300.00, and 360.00 μg/mL. CDR and DSM were obtained at 5.00, 50.00, 250.00, 400.00, 500.00, and 600.00 μg/mL. Samples were injected six times ($n = 6$). Calibration curves were subjected to linear regression analysis according to the equation $y = ax + b$, where y is the peak area, x is the sample concentration, a is the slope, and b is the y-intercept of the regression line. Linearity was established when the correlation coefficients ($r^2$) were >0.999.

The limits of detection (LODs) and limits of quantification (LOQs) were calculated using the standard deviations (SDs) of y-intercepts and slopes of calibration curves and the equations $LOD = 3.3 \times \sigma/S$ and $LOQ = 10 \times \sigma/S$ ($\sigma$ = SD of y-intercept, S = slope of the calibration curve).

Intra- and inter-day tests were performed at three concentrations (low, medium, and high) in linear ranges—i.e., L7GC at 10.00, 100.00, and 250.00 μg/mL; C7dGlu at 5.00, 50.00, and 125.00 μg/mL; and CDR and DSM at 12.50, 125.00, and 312.50 μg/mL. Intra-day precision was determined by analyzing three concentrations of each compound in quintuplicate in one day, and inter-day precision was measured by analyzing the same three concentrations five times ($n = 5$) on days 1, 3, and 5. Precisions were defined as relative standard deviations (%RSD), and %RSD was calculated using the standard deviation (SD)/mean $\times$ 100.

Analyte recoveries were calculated to confirm the percentage recoveries of analytes from *L. chinensis* extracts. Recovery experiments were performed by spiking *L. chinensis* extract with L7GC, CDR, C7dGlu, and DSM standards at three concentrations (low, medium, and high) and testing them in triplicate. Recovery % was calculated using (amount of analyte in spiked sample − amount of analyte in the sample)/amount of spiked standard $\times$ 100.

Solution stabilities of L7GC, CDR, C7dGlu, and DSM were checked by storing stock solutions at room temperature or 4 °C for 0, 6, 24, 48, and 72 h. The analysis was repeated five times ($n = 5$), and the % difference in the areas of each peak in the obtained chromatograms was calculated.

## 3. Results and Discussion

### 3.1. Development of HPLC-PDA analysis conditions

To develop a simultaneous analysis method for the L7GC, CDR, C7dGlu, and DSM of *L. chinensis* using HPLC-PDA, the solvent composition ratio (e.g., 0.1% formic acid–water/0.1% formic acid–acetonitrile, and 0.1% trifluoroacetic acid–water/0.1% trifluoroacetic acid-acetonitrile), column temperature (e.g., 25, 30, 35, and 40 °C), and flow rate (e.g., 0.5, 0.7, 0.8, and 1.0 mL/min) of the mobile phase were optimized. As a result, the four analyte peaks were separated under optimized conditions, and the total area of the four separated peak areas was confirmed to be at least 50% or more of the sum of the total

peak areas. Retention times were as follows: L7GC, 7.445 min; CDR, 9.445 min; C7dGlu, 10.121 min; and DSM, 15.034 min (Figure 2).

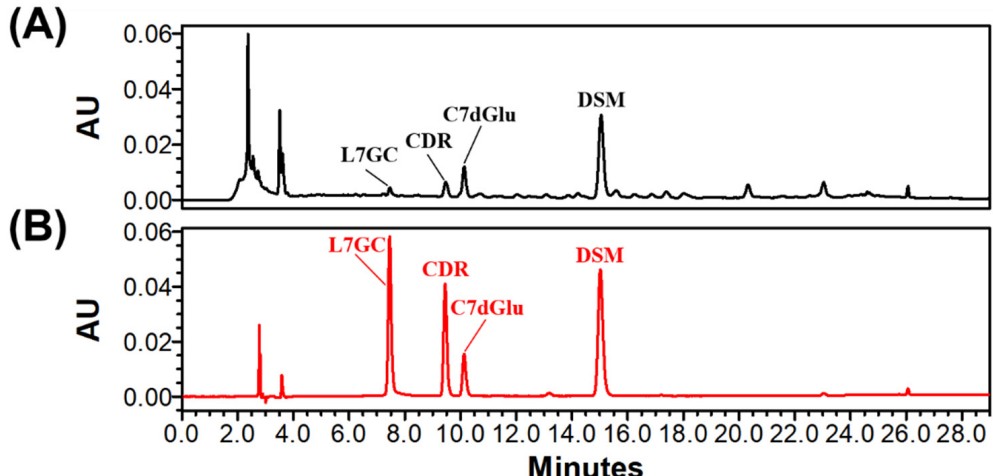

**Figure 2.** Chromatogram of *L. chinensis* extract at 340 nm (**A**); chromatogram of the standard mixture of L7GC, CDR, C7dGlu, and DSM at 340 nm (**B**).

### 3.2. HPLC-PDA Method Validation

#### 3.2.1. Specificity

The specificity was evaluated by confirming the retention times ($t_R$) and absorption spectra from HPLC-PDA chromatograms. For *L. chinensis* extract and the standard solutions, the retention times ($t_R$) were 7.445 min for L7GC, 9.445 min for CDR, 10.121 min for C7dGlu, and 15.034 min for DSM. In addition, a comparative analysis of the absorption spectra was performed in the range 190–400 nm. The PDA absorption maxima ($\lambda_{max}$) were the same at 254.4 nm and 347.4 for L7GC, 266.2 nm and 336.6 nm for CDR, 252.0 nm and 347.4 nm for C7dGlu, 252.0 nm and 346.2 nm for DSM. Thus, we chose an absorption wavelength of 340 nm for the simultaneous analysis of L7GC, CDR, C7dGlu, and DSM (Figure 3).

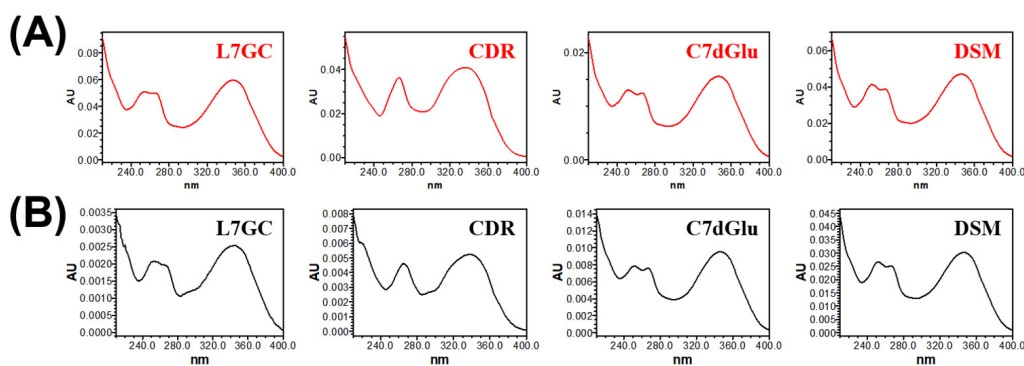

**Figure 3.** PDA spectra of the standard mixture (**A**) and *L. chinensis* extract (**B**).

#### 3.2.2. Linear Range, Linearity, Limits of Detection (LODs), and Limits of Quantification (LOQs)

Linearity was analyzed in the concentration range 3.00–360.00 µg/mL for L7GC and C7dGlu and 5.00–600.00 µg/mL for CDR and DSM. The correlation coefficients ($r^2$) of all curves were >0.999, which showed that the established analysis range showed a good linearity. The LOD and LOQ results were 2.152 and 6.521 µg/mL (for L7GC), 1.927 and 5.840 µg/mL (for CDR), 0.379 and 1.147 µg/mL (for C7dGlu), and 0.901 and 2.732 µg/mL (for DSM), respectively. These results suggest that the minimum detection concentrations of L7GC, CDR, C7dGlu, and DSM in *L. chinensis* extract lay in the range 0.379 to 2.152 µg/mL

and that quantitative analysis was possible from 1.147 µg/mL. The results are summarized in Table 1.

**Table 1.** Retention times, linear ranges, regression equations, coefficients of determination ($r^2$), limits of detection (LODs), and limits of quantitation (LOQs) of L7GC, CDR, C7dGlu, and DSM for the developed HPLC-PDA method (*n* = 6).

| Compound [a] | Retention Time (min) | Linear Range (µg/mL) | Regression Equation [b] | $r^2$ | LOD (µg/mL) | LOQ (µg/mL) |
|---|---|---|---|---|---|---|
| L7GC | 7.445 ± 0.02 | 3.00–360.00 | y = 11,715.29x − 5916.88 | 1.0000 | 2.152 | 6.521 |
| CDR | 9.445 ± 0.01 | 5.00–600.00 | y = 5316.01x − 10,240.04 | 0.9999 | 1.927 | 5.840 |
| C7dGlu | 10.121 ± 0.02 | 3.00–360.00 | y = 5487.39x − 15,317.53 | 0.9996 | 0.379 | 1.147 |
| DSM | 15.034 ± 0.03 | 5.00–600.00 | y = 1783.97x − 12,666.06 | 0.9991 | 0.901 | 2.732 |

[a] Luteolin-7-*O*-β-D-glucuronopyranosyl (1→2)-*O*-β-D-glucuronopyranoside, L7GC; clerodendrin, CDR; chrysoeriol-7-*O*-diglucuronide, C7dGlu; diosmin, DSM. [b] y = ax + b, y: peak area (AU), x: concentration (µg/mL).

### 3.2.3. Intra-and Inter-Day Precisions

For intra-day measurements of L7GC at 10.00, 100.00, and 250.00 µg/mL; C7dGlu at 5.00, 50.00, and 125.00 µg/mL; and CDR and DSM at 12.50, 125.00, and 312.50 µg/mL produced relative standard deviations (%RSD) of 0.11–1.93%. Similarly, inter-day measurements of L7GC at 10.00, 100.00, and 250.00 µg/mL; C7dGlu at 5.00, 50.00, and 125.00 µg/mL; and CDR and DSM at 12.50, 125.00, and 312.50 µg/mL produced relative standard deviations (%RSD) of 0.63–1.95%. The %RSD values for the intra- and inter-day precisions were all <2%, indicating a good precision and accuracy (Table 2).

**Table 2.** Inter- and intra-day precision of L7GC, CDR, C7dGlu, and DSM at low, medium, and high concentrations for the developed HPLC-PDA method.

| Compound [a] | Conc. (µg/mL) | Precision | | | |
|---|---|---|---|---|---|
| | | Intra-Day (*n* = 5) | | Inter-Day (*n* = 5) | |
| | | Measured Conc. (µg/mL) | RSD (%) | Measured Conc. (µg/mL) | RSD (%) |
| L7GC | 10.00 | 8.26 | 0.42 | 8.19 | 1.31 |
| | 100.00 | 100.46 | 0.20 | 97.95 | 1.96 |
| | 250.00 | 244.08 | 0.18 | 245.49 | 0.87 |
| CDR | 12.50 | 12.28 | 0.56 | 12.32 | 1.69 |
| | 125.00 | 119.56 | 0.54 | 118.95 | 0.77 |
| | 312.50 | 300.17 | 0.11 | 297.71 | 0.63 |
| C7dGlu | 5.00 | 5.06 | 0.92 | 5.08 | 1.29 |
| | 50.00 | 48.74 | 0.13 | 49.27 | 1.41 |
| | 125.00 | 122.60 | 0.32 | 123.86 | 1.65 |
| DSM | 12.50 | 11.33 | 1.93 | 11.35 | 1.41 |
| | 125.00 | 123.97 | 0.24 | 125.73 | 1.17 |
| | 312.50 | 305.70 | 1.02 | 301.60 | 1.95 |

[a] Luteolin-7-*O*-β-D-glucuronopyranosyl (1→2)-*O*-β-D-glucuronopyranoside, L7GC; clerodendrin, CDR; chrysoeriol-7-*O*-diglucuronide, C7dGlu; diosmin, DSM.

### 3.2.4. Analyte Recoveries

The recovery results for L7GC, CDR, C7dGlu, and DSM using the standard addition method were as follows. For C7GC, the recoveries from the 5.00, 50.00, and 125.00 µg/mL solutions were 106.04%, 99.24%, and 105.62%, respectively; the CDR recoveries from the 6.25, 62.50, and 156.25 µg/mL solutions were 113.18%, 99.55%, and 104.80%, respectively; the C7dGlu recoveries from the 5.00, 50.00, and 125.00 µg/mL solutions were 119.35%, 96.83%, and 104.01%, respectively; and the DSM recoveries from the 6.25, 62.50, and 156.25 µg/mL solutions were 127.07%, 103.07%, and 109.68%, respectively. The %RSD values for recovery ranged from 0.14 to 1.73% (Table 3).

**Table 3.** Recoveries of L7GC, CDR, C7dGlu, and DSM for the developed HPLC-PDA method (*n* = 3).

| Compound [a] | Original Conc. (μg/mL) | Spike Conc. (μg/mL) | Found Conc. (μg/mL) | Recovery [b] ± SD (%) | RSD (%) |
|---|---|---|---|---|---|
| L7GC | 0.81 | 5.00 | 5.70 | 106.04 ± 1.29 | 1.22 |
|  |  | 50.00 | 49.89 | 99.24 ± 0.35 | 0.35 |
|  |  | 125.00 | 130.02 | 105.62 ± 0.18 | 0.17 |
| CDR | 4.16 | 6.25 | 10.42 | 113.18 ± 0.87 | 0.77 |
|  |  | 62.5 | 64.63 | 99.55 ± 0.54 | 0.54 |
|  |  | 156.25 | 161.07 | 104.80 ± 0.23 | 0.22 |
| C7dGlu | 18.81 | 5.00 | 23.75 | 119.35 ± 1.84 | 1.54 |
|  |  | 50.00 | 67.44 | 96.83 ± 0.84 | 0.86 |
|  |  | 125.00 | 145.75 | 104.01 ± 0.21 | 0.20 |
| DSM | 22.78 | 6.25 | 29.06 | 127.07 ± 2.20 | 1.73 |
|  |  | 62.50 | 87.85 | 103.07 ± 0.86 | 0.83 |
|  |  | 156.25 | 189.95 | 109.68 ± 0.15 | 0.14 |

[a] Luteolin-7-*O*-β-D-glucuronopyranosyl (1→2)-*O*-β-D-glucuronopyranoside, L7GC; clerodendrin, CDR; chrysoeriol-7-*O*-diglucuronide, C7dGlu; diosmin, DSM. [b] Recovery (%) = (amount of analyte in spiked sample − amount of analyte in the sample)/amount of spiked standard × 100.

### 3.2.5. Stabilities of Analyte Solutions

The stabilities of the analytes in solution were evaluated by calculating the %RSD and %difference for the peak areas of L7GC, CDR, C7dGlu, and DSM after storage at room temperature or 4 °C for 0, 6, 24, 48, and 72 h. The peak areas of L7GC, CDR, C7dGlu, and DSM in solution decreased over time at both temperatures. The %difference values of the four compounds after 72 h were as follows: L7GC, −10.89% (R.T.) and −7.65% (4 °C); CDR, −5.65% (R.T.) and −6.15% (4 °C); C7dGlu, −5.80% (R.T.) and −7.86% (4 °C); DSM, −39.80% (R.T.) and −5.66% (4 °C). DSM showed a greater decrease in peak area over time at room temperature than the other compounds (RSD, 18.95%). In addition, at room temperature CDR showed a low RSD of 2.42%, while under refrigerated conditions DSM showed a low RSD of 2.13%. The %RSD values of the peak areas for the two conditions (room temperature or 4 °C) fell in the range of 2.42–18.95% for the room temperature condition and 2.13–3.66% for the 4 °C condition, respectively, indicating that solutions at both room temperature and 4 °C were stable below 18.95%. However, we recommend using sample solutions within 6 h of preparation (Table 4).

**Table 4.** Stabilities of L7GC, CDR, C7dGlu, and DSM in sample solutions at room temperature or 4 °C for 0, 6, 24, 48, and 72 h (*n* = 5).

| Compound [a] | Temp | Peak Area (Mean AU ± SD) | | | | | RSD (%) |
|---|---|---|---|---|---|---|---|
|  |  | 0 h | 6 h | 24 h | 48 h | 72 h |  |
| L7GC | R.T. | 19,437 ± 193 | 18,357 ± 155 | 18,510 ± 127 | 17,857 ± 92 | 17,321 ± 238 | 4.03 |
|  | 4 °C | 19,437 ± 193 | 18,314 ± 136 | 18,051 ± 109 | 17,542 ± 183 | 17,951 ± 138 | 3.66 |
| CDR | R.T. | 51,346 ± 340 | 49,122 ± 424 | 48,587 ± 181 | 48,438 ± 195 | 48,443 ± 615 | 2.42 |
|  | 4 °C | 51,346 ± 340 | 48,789 ± 483 | 49,070 ± 334 | 48,710 ± 335 | 48,188 ± 270 | 2.38 |
| C7dGlu | R.T. | 95,923 ± 1816 | 90,583 ± 881 | 89,581 ± 441 | 88,643 ± 1548 | 90,358 ± 1121 | 3.12 |
|  | 4 °C | 95,923 ± 1816 | 91,128 ± 650 | 89,458 ± 1438 | 88,591 ± 663 | 88,384 ± 830 | 3.35 |
| DSM | R.T. | 357,036 ± 1145 | 340,928 ± 1062 | 324,260 ± 5259 | 250,171 ± 4572 | 214,936 ± 637 | 18.95 |
|  | 4 °C | 357,036 ± 1145 | 339,828 ± 1173 | 341,246 ± 1379 | 341,845 ± 1559 | 336,843 ± 1513 | 2.13 |

[a] Luteolin-7-*O*-β-D-glucuronopyranosyl (1→2)-*O*-β-D-glucuronopyranoside, L7GC; clerodendrin, CDR; chrysoeriol-7-*O*-diglucuronide, C7dGlu; diosmin, DSM.

### 3.3. Quantification of Four Marker Compounds in L. chinensis Extract

The contents of four compounds in *L. chinensis* were calculated from the corresponding calibration curves and the precision was found to be less than 2%. As a result, the contents of L7GC, CDR, C7dGlu, and DSM in the *L. chinensis* extract were 0.11 ± 0.001 mg/g, 0.58 ± 0.003 mg/g, 1.01 ± 0.017 mg/g, and 10.36 ± 0.032 mg/g, respectively. Out of four compounds, DSM was the most abundant marker compound in *L. chinensis* extract (Table 5).

**Table 5.** Contents of L7GC, CDR, C7dGlu, and DSM marker compounds in *L. chinensis* extract sample ($n = 5$).

| Compound [a] | Contents (mg/g) | |
|---|---|---|
| | Mean ± SD | RSD (%) |
| L7GC | 0.108 ± 0.001 | 0.76 |
| CDR | 0.579 ± 0.003 | 0.55 |
| C7dGlu | 1.014 ± 0.017 | 1.63 |
| DSM | 10.362 ± 0.032 | 0.31 |

[a] Luteolin-7-*O*-β-D-glucuronopyranosyl (1→2)-*O*-β-D-glucuronopyranoside, L7GC; clerodendrin, CDR; chrysoeriol-7-*O*-diglucuronide, C7dGlu; diosmin, DSM.

### 4. Conclusions

In this study, the HPLC-PDA analysis method was developed for the simultaneous determination of four marker compounds (luteolin-7-*O*-β-D-glucuronopyranosyl (1→2)-*O*-β-D-glucuronopyranoside, clerodendrin, chrysoeriol-7-*O*-diglucuronide, and diosmin) in *L. chinensis* extract. The developed analytical method was validated by determining its specificities, linearities, limits of detection (LODs), limits of quantification (LOQs), precisions, analyte recoveries, and solution stabilities. The simultaneous measurement method devised in the present study is believed to be suitable for quality control and standardization studies on *L. chinensis* and related plant species.

**Author Contributions:** Conceptualization, S.-N.K. and M.H.Y.; investigation, B.-G.J. and Y.-H.P.; data curation B.-G.J., Y.-H.P. and K.H.K.; writing—original draft preparation B.-G.J. and Y.-H.P.; writing—review and editing, S.-N.K. and M.H.Y. All authors have read and agreed to the published version of the manuscript.

**Funding:** This research was supported by the Bio and Medical Technology Development Program of the National Research Foundation (NRF) funded by the Ministry of Science and ICT (NRF-2019M3A9I3080263, NRF-2019M3A9I3080265, and NRF-2019M3A9I3080266).

**Institutional Review Board Statement:** Not applicable.

**Informed Consent Statement:** Not applicable.

**Data Availability Statement:** The data presented in this study are available on request from the corresponding author.

**Conflicts of Interest:** The authors declare no conflict of interest.

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
