# Peer review of "Simultaneous Determination of Four Marker Compounds in Lobelia chinensis Lour. Extract by HPLC-PDA"

_applsci, doi:10.3390/app112412080_

Round 1
Reviewer 1 Report
The presented study deals with development and validation of a fairly simple RP-HPLC/UV-Vis method for detection of selected bioactive compounds in Lobelia chinensis plant. The method was validated using standard solutions (and the plant material for the recovery study) regarding typical parameters such as linearity, LOD, LOQ, etc.
My remarks follow:
From a screening research on the topic of L. chinensis analysis using HPLC, two studies popped up and since they are not included in the cited literature, I would like the authors to discuss them in relation to their presented topic. These would be:
1) Sensen Li et al., An on-line high-performance liquid chromatography−diode-array detector−multi-stage mass spectrometry−deoxyribonucleic acid−4′,6-diamidino-2-phenylindole−fluorescence detector system for screening the DNA-binding active compounds in Fufang Banbianlian Injection, Journal of Chromatography A, 1424 (2015) 37–50 (http://dx.doi.org/10.1016/j.chroma.2015.10.079), and
2) Sensen Li et al., Rapid Identification and Assignation of the Active Ingredients in Fufang Banbianlian Injection Using HPLC-DAD-ESI-IT-TOF-MS, Journal of Chromatographic Science, 54 (2016) 1225–1237 (https://doi.org/10.1093/chromsci/bmw055).
line 106: What is meant by "10 mL of 10-fold (w/v) distilled water"? Distilled water should not have any (w/v) content of anything...
lines 116 and 117: "DMSO-80% aqueous methanol (1:1, v/v)" - what is meant by this? Is this 80% solution of DMSO in methanol/water 1:1? Or 1:1 solution of DMSO in 80% aqueous methanol? Please rephrase to clarify.
line 144: Given that similar linearity ranges were established for L7GC and C7dGlu (3-360 ug/mL), why were different concentrations used for the precision analysis of these two compounds?
line 154: "amount analyte in spiked sample" and "amount analyte in sample" should be changed to "amount of analyte in the spiked sample" and "amount of analyte in the sample", respectively.
line 161: Presentation of charts describing changes in separation efficiency (expressed as e.g. resolution of target peaks) based on the variation of optimized parameters (i.e. solvent composition ratio, column temperature, and flow rate) could be useful for the purpose of adoption of this method on HPLC systems with limited flexibility (e.g. regarding column heating). In other words - the individual contribution of these three factors could be provided (at least in the Supporting Information).
line 188: The sentence could be shorted as effectively only two ranges were used (i.e. the same for L7GC and C7dGlu, or CDR and DSM, respectively).
Figure 4: The images are too small in my opinion. Is the information "(n=6)" related to the number of measurements for each calibration point? If so, please state this explicitly in the figure caption.
line 195: I suggest to rephrase the sentence to: "LOD and LOQ results were 2.152 and 6.521 μg/mL (for L7GC), 1.927 and 5.840 μg/mL (for CDR), 0.379 and 1.147 μg/mL (for C7dGlu), and 0.901 and 2.732 μg/mL (for DSM), respectively."
line 203: There are different fonts used in the table legend - please unify them (maybe this issue should be addressed by the editor's team).
Table 2: What procedure was used for accuracy calculation? The details are not mentioned neither in the Discussion nor in the Experimental section of the manuscript. Also, what is the purpose of the "Spiked amount" column in the table? If it is related to the accuracy measruement procedure, it should be explained. Lastly, there is a line under the the first data row that I suppose should not be there. Please remove it.
Table 3: I cannot obtain the values for Recovery stated in the table, if I try to use the Spike Concentration and Found Concentration inputs to calculate it. Can you please explain it better? I tried to follow the formula in the Experimental, but I think it's impossible since you are giving only two values and the formula asks for three (i.e. "amount analyte in spiked sample", amount analyte in sample" and "amount of spiked standard").
Table 4: How is the final RSD calculated? Is it obtained from the values of mean peak areas from individual time points? I think in this case the decrease in the peak are should be presented as a precentage. For example: a decrease from 19437.4 to 17320.8 AU for L7GC (0h, and 72h in R.T., respectively) presents a 10.89% drop in the peak area, which (in my opinion) is a much more comprehensible interpretation of the obtained data. Can you please explain the reason for using %RSD and also the interpretation that the reader should use for such kind of presented output (e.g. "4.03% RSD for L7GC at R.T. after 72h" compared for example to "a 10.89% drop in peak area for L7GC at R.T. after 72h")?
Since the topic of the manuscript is develompent of a method for evaluation of L. chinensis extracts and an extract was prepared for the determination of specificities and recoveries of target analytes, I assumed the study would end with the analysis of the extract and presentation of the measured values. Why this was not included in the study as a proof-of-concept for the developed method? It would be highly beneficial to add such data in my opinion.
Author Response
Please see an attached file.

Reviewer 2 Report
The manuscript presented by the authors, develops a methodology for the analysis of 4 flavonoids in Lobelia chinensis by HPLC-PDA at 340 nm. According to the HPLC-PDA chromatographic profile in figure 2, these are the priority compounds at this wavelength. However, the authors could make adjustments for the proposed work:
- The flavonoids in figure 1, can be grouped into a single flavonoid structure using R, R1 and R2 as substituents.
- They could exclude the "As a result, an Aegispak C18-L reversed-phase col- 164 umn (4.6 mm × 250 mm i.d., 5 μm) at 40°C and gradient elution using 0.1% formic acid- 165 water and 0.1% formic acid-acetonitrile were used as mobile phases at an optimized flow 166 rate of 1.0 mL/min" from results and discussion, since they already have this data in materia.
- According to the data described in the methodology, for each standard 6 different concentrations were prepared, and therefore n=6. However, in Figure 4 , each graph has only 5 points.
- Review tables 2 and 3, you have additional rows that should be extracted.
- The title of the article says "Simultaneous Determination of Four Marker Compounds in 2 Lobelia chinensis Extract by HPLC-PDA", however little is said about the determination of these flavonoids in the extract. Either change the text, or include some discussion in the text.
The author should present the quantification of these flavonoids in the extract
Author Response
Please see an attached file.

Reviewer 3 Report
The research presented in the manuscript is focused on the development of the HPLC-PDA method for simultaneous determination of four naturally occurring compounds in the herb Lobelia chinensis Lour.
In my opinion the manuscript, in the present form, is not appropriate for publication.
There are following remarks that should be addressed by the authors.
- In general, the style and precision of the language should be improved and repetition of text should be avoided.
- Introduction: other reported studies on L. chinensis should be cited and shortly described.
- 1: the abbreviations of the compound names should be also placed at their structures.
- Section 2.4: the repeated information on the used column is not necessary. There is also not needed to repeat “linear gradient” at each step of elution.
- Section 2.7: the title should be replaced by “Validation of the HPLC-PDA method”
- Section 2.7: in general the determination of validation parameters was not clearly presented both in terms of content and style of the language, e.g.
- what does it mean that “To assess linearity, the stock solutions … were diluted with six target concentration ranges (1%, 10%, …) to obtain …”.
- “Calibration curves were subjected to linear regression analysis…” or rather the obtained results were subjected to linear regression analysis, the further statements in this paragraph are also not clearly written
- what does it mean “ standard deviation of responses” (which responses – it should be better specified)
- “Precision analysis was used to check analyte stability” – “precision analysis” is not used to estimate analyte stability. Analyte stability is usually estimated as the recovery determined for test solutions/extracts in relation to the freshly prepared solution/extracts
- “Intra-day precision was determined by analyzing three concentrations (or rather samples with three concentrations of each compound ?) …”
- There is lack of description how the accuracy of the method was determined (Table 2). Besides that, accuracy should be rather expressed in % as a bias
- What does it mean the last sentence in this section: “…, and stabilities were evaluated by comparing the HPLC-PDA peak areas with baseline values” ?
7. Section 3.1: the content of this section is inadequate to the title, because the optimization was not described and only some analytical parameters considering in this optimization were listed.
8. Fig. 2: “HPLC-PDA chromatogram” should be replaced by “Chromatogram”
9. Fig. 4: this figure presenting calibration graphs is not necessary, presentation of regression equations in Table 1 is enough.
10. Table 4: taking into account the standard deviation for repeated measurements, the values of the peak areas are given too precisely.
11. Conclusion: this section should not be written as an abstract. In this section the conclusions driven from the obtained results and some important features of the developed method, as well as its future potential applications should be emphasized.
Author Response
Please see an attached file.

Round 2
Reviewer 1 Report
My remarks to the second version of the manuscript:
1) The notation of lines is different in the new pdf, as for example correction of "10 mL of 10-fold (w/v) distilled water" to "10 mL distilled water" is supposed to be present on line 108 while i see it on line 132 in the pdf of "manuscript v2". Furthermore all the corrections are highlighted together with the parts to be removed, which makes the new version hard do read. Therefore I am not able to assess whether the citations 32 and 33 were added with the appropriate comments (all occurences of these citations seem to be "crossed out" in the current version of the manuscript).
2) "Both compounds (L7GC and C7dGlu) were isolated from L. chinensis extract and used for this study. We obtained different amount of those compounds from L. chinensis, thus used different concentrations for the precision analysis. The precision analysis was performed in accordance with the ICH guideline and we measured three different concentrations of two compounds within the linearity range."
- In the Experimental part it is written that those compounds were isolated. I would assume this means dry form was obtained for further evaluation. In such a case, any concentration could be obtained.
3) Q6 - I think the authors answered a different question than I asked. Nevertheless, this is a minor thing.
4) Q10 - The authors claim that the fonts are unified while the opposite is true (look at the beta symbols) - at least in the version I am looking at. It seems wrong to me that the authors are making an obviously untrue statement.
5) Q12 - The numbers used for the calculation are not presented in the table. They should be provided to prove tahat the calculations were done right. Also, how was the amount of analyte in sample before spike determined?
6) Q13 - I still do not understand the point of %RSD determination across all timepoints, but if the % differences were added as well, it can probably stay there. However, in the present form I am not able to see where this was added since the line numbers don't match (as I mentioned earlier).
In the present form, some parts of the manuscript should still be improved. I would be glad to revise the manuscript once more with the revisions accepted so that I can better see what text was left (or added) and so that I can better navigate regarding the line numbers to see whether certain earlier remarks were addressed (it would be ideal if the authors pointed the correct line numbers the next time).
Author Response
Reviewer 1
My remarks to the second version of the manuscript:
1) The notation of lines is different in the new pdf, as for example correction of “10 mL of 10-fold (w/v) distilled water” to “10 mL distilled water” is supposed to be present on line 108 while I see it on line 132 in the pdf of “manuscript v2”. Furthermore, all the corrections are highlighted together with the parts to be removed, which makes the new version hard do read. Therefore, I am not able to assess whether the citations 32 and 33 were added with the appropriate comments (all occurrences of these citations seem to be “crossed out” in the current version of the manuscript).
- In the revised version of manuscript, “10 mL of 10-fold (w/v) distilled water” is changed to “10 mL distilled water” as follows: line 113, and citations 32 and 33 were added with appropriate comments as follows: line 57-62.
2) “Both compound (L7GC and C7dGlu) were isolated from L. chinensis extract and used for this study. We obtained different amount of those compounds from L. chinensis, thus used different concentrations for the precision analysis. The precision analysis was performed in accordance with the ICH guideline and we measured three different concentrations of two compounds within the linearity range.” - In the Experimental part it is written that those compounds were isolated. I would assume this means dry form was obtained for further evaluation. In such a case, any concentration could be obtained.
- L7GC (10.5 mg) and C7dGlu (5.2 mg) were isolated by MPLC flash chromatography using MeOH-water (1:9 → 2:8 → 3:7 → 100% MeOH) as eluent. Those compounds were further purified by preparative HPLC (ACN : 0.1% formic acid in water = 17:83) to afford L7GC (3.1 mg, purity ≥ 98.0%) and C7dGlu (1.8 mg, purity ≥ 98.0%). As a result, the amount of C7dGlu was less than L7GC, so that different concentrations were used for precision analysis.
3) Q6 - I think the authors answered a different question than I asked. Nevertheless, this is a minor thing.
- Thank you.
4) Q10 - The authors claim that the fonts are unified while the opposite is true (loot at the beta symbols) – at least in the version I am looking at. It seems wrong to me that the authors are making an obviously untrue statement.
- We carefully double-checked the manuscript, but couldn’t find any difference. As suggested by reviewer #1, this issue should be addressed by the editor's team.
5) Q12 - The numbers used for the calculation are not presented in the table. They should be provided to prove that the calculations were done right. Also, how was the amount of analyte in sample before spike determined?
- The amount of analyte in sample before the spike was calculated using a calibration curve obtained by the linearity test at a concentration of 10 mg/mL. It is newly added to Table 3 in the revised manuscript (line 226-228).
6) Q13 - I still do not understand the point of %RSD determination across all timepoints, but if the % differences were added as well, it can probably stay there. However, in the present from I am not able to see there this was added since the line numbers don’t match (as I mentioned earlier).
- % difference values are newly added to the revised version of manuscript and the line numbers now match (line 233-236).
Reviewer 3 Report
The article was partly improved, however there are still some issues which should be completed and corrected.
Q2. Introduction: other reported studies on L. chinensis should be cited and shortly described.
To show a novelty of the presented work, some more detailed information about the cited articles [32, 33] should be added, especially related to quantitative analysis of L. chinensis, e.g. how many and which compounds were determined, which methods were used, etc.
Q13. Section 3.1: the content of this section is inadequate to the title, because the optimization was not described and only some analytical parameters considering in this optimization were listed.
Information on which optimization method was applied should be added (e.g. method of one independent variable?), as well as some optimization results should be given, e.g. as the Supplementary materials.
Q16. Table 4: taking into account the standard deviation for repeated measurements, the values of the peak areas are given too precisely.
I understand that a peak area value is the average value of several replicate measurements, but in any case when the determined measurement error is in the order of hundreds (in any units), it makes no sense to enter its result with an accuracy of hundredth or tenths. Thus, the results should be given as follows: 19437±193; 18357±155; 18510±127 etc.
Author Response
Reviewer 3
The article was partly improved, however there are still some issues which should be completed and corrected.
Q2. Introduction: other reported studies on L. chinensis should be cited and shortly described.
To show a novelty of the presented work, some more detailed information about the cited articles [32, 33] should be added, especially related to quantitative analysis of L. chinensis, e.g. how many and which compounds were determined, which methods were used, etc.
- It is now newly added to the Introduction part as suggested by reviewer (line 57-62).
Q13. Section 3.1: the content of this section is inadequate to the title, because the optimization was not described and only some analytical parameters considering in this optimization were listed.
Information on which optimization method was applied should be added (e.g. method of one independent variable?), as well as some optimization results should be given, e.g. as the Supplementary materials.
- We agree with the reviewer's opinion, and “Optimization of HPLC-PDA analysis conditions” is now changed to “Development of HPLC-PDA analysis conditions” (Section 3.1).
Q16. Table 4: taking into account the standard deviation for repeated measurements, the values of the peak areas are given too precisely.
I understand that a peak area value is the average value of several replicate measurements, but in any case, when the determined measurement error is in the order of hundreds (in any units), it makes no sense to enter its result with an accuracy of hundredth or tenths. Thus, the results should be given as follows: 19437±193; 18357±155; 18510±127 etc.
- It is now changed as suggested by reviewer (Table 4).
Round 3
Reviewer 1 Report
Figure 1: A typo: "chrysoe-riol" - should be probably just "chrysoeriol"
Table 3: Recoveries don't match the formula that the authors provided in the response to the first report (and are also providing on lines 160-161). For example the first row: (5.70-0.81)/5×100 = 97.8. Is there an error or am I calculating the values in a wrong way?
In general, the headers of invidividual columns in all tables should be explained better in the corresponding table legends.
line 234: "The %difference values of the four compounds were less than 10.89% and 9.75% for L7GC ..." In my opinion the sentence should say they "were 10.89% and 9.75% for L7GC" (and correspondingly for other compounds). Or it should be stated that this was the lowest value after 72h, but nevertheless the values reached those numbers, so it makes no sense to say "less than".
Table 5: The RSD values don't match the results (i.e. for 0.11±0.001 the %RSD should be 0.91%). If more precise numbers were used for the calculation, they should be provided - on top of that, there should be the same number of decimal places in the mean and SD values, so I suggest to include 3 decimal places for the means in Table 5.
Author Response
Reviewer 1
Q1. Figure 1: A typo: "chrysoe-riol" - should be probably just "chrysoeriol"
- "chrysoe-riol" is now changed to "chrysoeriol" (line 94).
Q2. Table 3: Recoveries don't match the formula that the authors provided in the response to the first report (and are also providing on lines 160-161). For example the first row: (5.70-0.81)/5×100 = 97.8. Is there an error or am I calculating the values in a wrong way?
- The first row: (5.701-0.806)/4.616×100 = 106.04.
Q3. In general, the headers of invidividual columns in all tables should be explained better in the corresponding table legends.
- It is now changed and please see the red-colored parts in the table legends.
Q4. line 234: "The %difference values of the four compounds were less than 10.89% and 9.75% for L7GC ..." In my opinion the sentence should say they "were 10.89% and 9.75% for L7GC" (and correspondingly for other compounds). Or it should be stated that this was the lowest value after 72h, but nevertheless the values reached those numbers, so it makes no sense to say "less than".
- It is now corrected to ‘The %difference values of the four compounds after 72 h were as follows: L7GC, -10.89% (R.T.) and -7.65% (4°C); CDR, -5.65% (R.T.) and -6.15% (4°C); C7dGlu, -5.80% (R.T.) and -7.86% (4°C); DSM, -39.80% (R.T.) and -5.66% (4°C).’ (line 235-238)
Q5. Table 5: The RSD values don't match the results (i.e. for 0.11±0.001 the %RSD should be 0.91%). If more precise numbers were used for the calculation, they should be provided - on top of that, there should be the same number of decimal places in the mean and SD values, so I suggest to include 3 decimal places for the means in Table 5.
- Mean ± SD in Table 5 are now changed as suggested by reviewer.